# Three-Source Partitioning of Methane Emissions from Paddy Soil: Linkage to Methanogenic Community Structure

**DOI:** 10.3390/ijms20071586

**Published:** 2019-03-29

**Authors:** Jing Yuan, Xiaomei Yi, Linkui Cao

**Affiliations:** 1School of Agriculture and Biology, Shanghai Jiao Tong University, 800 Dongchuan Road, Shanghai 200240, China; yuanjing1203@zju.edu.cn (J.Y.); yixiaomei@sjtu.edu.cn (X.Y.); 2Institute of Soil and Water Resources and Environmental Science, College of Environmental and Resource Sciences, Zhejiang University, Hangzhou 310058, China

**Keywords:** organic fertilizer, methane, methanogen, paddy, ^13^C

## Abstract

Identification of the carbon (C) sources of methane (CH_4_) and methanogenic community structures after organic fertilization may provide a better understanding of the mechanism that regulate CH_4_ emissions from paddy soils. Based on our previous field study, a pot experiment with isotopic ^13^C labelling was designed in this study. The objective was to investigate the main C sources for CH_4_ emissions and the key environmental factor with the application of organic fertilizer in paddies. Results indicated that 28.6%, 64.5%, 0.4%, and 6.5% of ^13^C was respectively distributed in CO_2_, the plants, soil, and CH_4_ at the rice tillering stage. In total, organically fertilized paddy soil emitted 3.51 kg·CH_4_ ha^−1^ vs. 2.00 kg·CH_4_ ha^−1^ for the no fertilizer treatment. Maximum CH_4_ fluxes from organically fertilized (0.46 mg·m^−2^·h^−1^) and non-fertilized (0.16 mg·m^−2^·h^−1^) soils occurred on day 30 (tillering stage). The total percentage of CH_4_ emissions derived from rice photosynthesis C was 49%, organic fertilizer C < 0.34%, and native soil C > 51%. Therefore, the increased CH_4_ emissions from paddy soil after organic fertilization were mainly derived from native soil and photosynthesis. The 16S rRNA sequencing showed *Methanosarcina* (64%) was the dominant methanogen in paddy soil. Organic fertilization increased the relative abundance of *Methanosarcina*, especially in rhizosphere. Additionally, *Methanosarcina* sp. 795 and *Methanosarcina* sp. 1H1 co-occurred with *Methanobrevibacter* sp. AbM23, *Methanoculleus* sp. 25XMc2, *Methanosaeta* sp. HA, and *Methanobacterium* sp. MB1. The increased CH_4_ fluxes and labile methanogenic community structure in organically fertilized rice soil were primarily due to the increased soil C, nitrogen, potassium, phosphate, and acetate. These results highlight the contributions of native soil- and photosynthesis-derived C in paddy soil CH_4_ emissions, and provide basis for more complex investigations of the pathways involved in ecosystem CH_4_ processes.

## 1. Introduction

Climate change is mainly attributed to rapidly rising atmospheric concentrations of greenhouse gases (GHGs) such as carbon dioxide (CO_2_), H_2_O, methane (CH_4_) and nitrous oxide (N_2_O) [1]. The atmospheric concentrations of CH_4_, the GHGs present at the third-highest concentration (after CO_2_ and H_2_O), are influenced by anthropogenic activities. The global warming potential of CH_4_ on a 100-year time horizon is 28–34 times higher than CO_2_ [2]. Approximately 20% of the global warming that occurred after industrialization has been attributed to atmospheric CH_4_ emissions [3]. Paddy soils are the largest anthropogenic source of CH_4_, accounting for approximately 10% of global CH_4_ emissions [4]. Emissions of CH_4_ from paddy soils are concentrated in irrigated areas; irrigated paddy soils account for 60% of the total rice harvesting area worldwide, but produce 78% of CH_4_ emissions in rice-producing areas [3,5]. Several factors are known to affect paddy CH_4_ emissions, including climate (temperature, precipitation, humidity), crops (cultivar, root vitality, aeration), soil characteristics (moisture, O_2_, Eh, pH, texture, organic matter, mineral nutrition), and field management (water management, fertilization, farming operations) [6,7,8,9,10]. Changes to any of these factors can alter the soil microenvironment and in turn affect overall CH_4_ emissions [11]. Especially, paddy fields treated with organic fertilizers release more CH_4_ than inorganically fertilized and unfertilized soils [12,13]. 

The total organic C budget in the paddy ecosystem includes C fixed through rice photosynthesis, as well as native organic C present in soil, rice plant residues and sediments [14]. Decomposition of organic matter by methanogens under anaerobic conditions leads to production of CH_4_. Overall CH_4_ emissions from paddy soil are the net result of CH_4_ production, oxidation and transportation [15]. There are two major pathways of CH_4_ production: acetoclastic and hydrogenotrophic (Figure 1). Acetoclastic methanogens [16,17] use ATP to convert acetate to acetyl phosphate and then remove the phosphate ion via a reaction catalyzed by coenzyme A. CH_4_ is formed gradually by processes involving oxidized ferredoxin, tetrahydrosarcinapterin, coenzyme M, and coenzyme B. In contrast, hydrogenotrophic methanogens [18] initially form formylmethane furans (CHOMFR) from H^+^ and methanofuran, then CH_4_ is finally generated via a series of complex catalytic reactions. Overall, the metabolic pathways by which methanogens produce CH_4_ involve numerous enzymes and require large quantities of electrons, C substrates and energy [19,20]. In most soils, CH_4_ is mainly (58–75%) produced from acetate, except in permanent pasture soil (42%) [21]. Hydrogenotrophic methanogenesis produces about 22–37% of CH_4_. Subsequently, the CH_4_ produced is partially consumed by methanotrophs via the phosphoribulose or serine pathways [22]; the remainder diffuses into the atmosphere through rice aerenchyma or via ebullition (bubbles of gas in water) [3,23]. When an organic fertilizer is applied to soil (Figure 1), some nutrients are assimilated by the plants, while others would be reserved within the soil and utilized by soil microorganisms. In comparison with chemical fertilization and no fertilization, organic fertilization improves soil quality by enhancing a series of complex microbial metabolic activities, such as sulfate reduction, iron reduction, nitrogen cycle, and carbon cycle [9,24,25]. Ultimately, organic fertilizers provide a large amount of energy and substrates to methanogens and, thus, stimulate production of CH_4_ [25]. This effect has been demonstrated in many other studies [26]. In addition, rice plant growth and photosynthesis are also improved by organic fertilization [27]. Consequently, more C-containing compounds are transported from the rice plant tissues into the soil, which eventually increases the availability of C substrates for CH_4_ production [9]. Therefore, controlling and limiting the availability of additional C sources to methanogens could potentially mitigate the increased CH_4_ emissions observed in organically-managed rice soils. 

Studies have emphasized that increased CH_4_ emissions from organically fertilized paddy soils are due to the extra C input provided by the organic fertilizer [27,28,29]. In our previous paddy field study [30,31,32], we applied four treatments with chemical fertilizer (CT), organic fertilizer (OT), mixed with chemical and organic fertilizer (MT), and no fertilizer (ctrl). Results found that CH_4_ emissions were completely higher in OT than that in MT, CT or ctrl. Moreover, soil nitrogen (N), C, phosphate (P), potassium (K), and acetate were the main key environmental factors associated with CH_4_ emissions and the dominant methanogenic communities. The hydrogenotrophic *Methanocella* and acetoclastic *Methanosaeta* were the predominant methanogenic communities. Especially, application of organic fertilizer promoted the dominant acetoclastic methanogens, but suppressed the dominant hydrogenotrophic methanogens. While, another study [25] found that the abundance of *Methanosarcina* and *Methanosaeta* at the rice tillering stage decreased rapidly after application of composted manure compared to non-composted manure treatment. Furthermore, a decrease in *Methanosarcina*, but increase in *Methanobrevibacter*, was observed in the composted treatment in contrast to the control treatment [33]. Varied fertilization could have varied effects on CH_4_ emissions and methanogens from paddy soil. The transformation in the methanogenic community structure and enhanced availability of C substrates could explain the increased CH_4_ production in OT compared to other treatments. However, the specific contribution of the additional C from organic fertilizer on paddy CH_4_ emissions are unclear and merit more assessment. We hypothesized there were three C sources responsible for the CH_4_ emissions in organically-fertilized paddy soil: rice photosynthesis-derived, native soil C-derived, and organic fertilizer-derived. 

The contribution of root-derived CO_2_ in paddy ecosystems indicates photosynthesis should be considered as one of the main drivers of soil C flow [14]. Photosynthesis-derived C (including rice residues, root tissues, root CO_2_, etc.) is assimilated by methanogenic archaea present on the rice roots [34]. The rates of photosynthesis-derived CH_4_ emissions from rice fields are controlled by the abundance of active methanogens on the rice roots [35]. While, the native soil C is also a large C source for CH_4-_producing methanogens. Before rice transplantation, paddy soils contain both labile and ‘old’ native C [8]. Different C substrates have stimulating or restraining effects, termed positive or negative priming effects, on the mineralization of native soil organic C [36]. Addition of substrates with relatively high C availability and nutrient contents often exerts a priming effect, and increases microbial activity [37]. The increased soil levels of such easily degraded compounds greatly enhance decomposition of native soil C [38,39]. After addition of organic fertilizer, old soil C becomes more labile and is more easily utilized by microbes. However, the role of old native soil C in the priming of soil organic C as a source of CH_4_ production is not well established. Moreover, we also assumed that the C derived from the organic fertilizer can also contribute to the increased CH_4_ emissions in paddy soils. However, it is complicated to separate the contributions of soil native C and organic fertilizer C due to the limitation of labelling approaches. Based on our former field research, we applied a pot experiment with isotopic ^13^C labelling in this study. Paddy CH_4_ and CO_2_ emissions, and some soil variables (N, C, P, K, and acetate) for the whole rice cultivation were firstly investigated. Besides, the relative abundances of methanogenic communities on the roots and in the soil, were measured by microbial 16S rRNA sequencing. Moreover, we calculated the relative contributions of rice photosynthesis-derived, native soil C-derived, and organic fertilizer-derived C to the C sources for CH_4_ emissions from paddy soil. The co-occurrence networks between methanogenic species and environmental factors were also analyzed. The objectives of this research were to elucidate: (1) the main C source(s) responsible for the increased CH_4_ emissions observed in rice soil after addition of organic fertilizer; (2) the dominant methanogenic community in organically fertilized paddy soils; and (3) the main factors affecting CH_4_ effluxes and methanogenic community structures in paddy soil.

## 2. Results

### 2.1. Percentage δ^13^C Fractionation

At the regreening stage (Figure 2a,b), CO_2_ accounted for 93.42% of δ^13^C fractionation on average in all samples, followed by the rice plants ^13^C (6.50%) and soil ^13^C (0.07%). At the regreening stage the δ^13^CH_4_ distribution was no greater than 0.01%. At the tillering stage, the percentages of δ^13^C fractionation in the CO_2_, rice plants, soil, and CH_4_ were 28.60%, 64.47%, 0.40%, and 6.53%, respectively. The proportion of δ^13^CH_4_ in the organic fertilizer treatment was 7.28%, higher than that in the no fertilizer treatment (5.78%). Ultimately, at the end of the booting stage, 86.49–91.38% of ^13^C was fixed within the rice plants; 0.62–1.08% was stored in the soil, and 2.19–2.98% was transformed and emitted in the form of ^13^CH_4_. Compared to the no fertilizer treatment, the addition of organic fertilizer increased CH_4_ emissions derived from rice photosynthesis. We calculated organic fertilizer directly accounted for, at most, 0.34% of the C sources of CH_4_ emissions (Figure 2c). Photosynthesis and native soil C accounted for at least 49.06% and 50.60% of the C sources of CH_4_ emissions, respectively. Hence, rice photosynthesis-derived and native soil-derived C flow were the two primary C sources of CH_4_ emissions. To analyze which C sources affected CH_4_ emissions most, contribution coefficients were generated using the following linear regression equations:δ^13^CH_4_ and rice plant δ^13^C: *y_r_* = 6.00*x* + 120.47, *K_r_* = 0.17(1)
δ^13^CH_4_ and rice soil δ^13^C: *y_s_* = 0.08*x* + 0.94, *K_s_* = 12.50 (2)

According to the linear regression analysis, daily δ^13^CH_4_ was significantly (*p* < 0.01) and positively dependent on both daily rice plant δ^13^C and soil δ^13^C (Figure 2d,e). *K_s_* was higher than *K_r_*, revealing that CH_4_ emissions were more sensitive to soil C fixation than plant photosynthesis C accumulation. 

### 2.2. Methanogenic Community Structures

After 16S rRNA sequencing, the differential abundance analysis of each sample was compared using volcano plots and a Venn plot (Figure 3a–d). More differences in the OTUs were observed between the non-root soil than root soil samples overall. The NMDS and RDA plots (Figure 3e,f) also revealed that microbial community structures in root soil samples differed from non-root soil samples, especially after organic fertilizer was applied. At the taxonomic class level (Figure 4a), *Euryarchaeota* (48.56%), *Crenarchaeota* (3.45%), *Thaumarchaeota* (1.54%), and other non-ranked archaeal communities (38.86%) were identified in all treatments. At the genus level (Figure 4b), eight genera of methanogens were identified: 63.89% *Methanosarcina*, 8.64% *Methanoculleus*, 6.42% *Methanobacterium*, 1.87% *Methanomethylovorans*, 0.06% *Methanosaeta*, 0.04% *Methanosphaera*, 0.38% *Methanobrevibacter*, 0.65% *Methanolobus*, and 40.50% unknown methanogens. *Methanosarcina* was the dominant methanogenic community. *Methanosarcina* had a higher relative abundance in O_root soil (87.75%) than the other samples. In addition, the relative abundance of *Methanoculleus*, the second largest methanogenic community, was higher in the no fertilizer treatment than organic fertilizer treatment. RDA (Figure 3f) and networks (Figure 5) were generated to analyze the correlations between the methanogenic species and environmental variables, and indicated the methanogenic community structures were susceptible to soil available phosphate (AP), available potassium (AK), available nitrogen (AN), soil C, CO_2_ efflux, rice C, and biomass. Moreover, *Methanobacterium palustre* co-existed with *Methanoculleus* sp. 25XMc2, *Methanosaeta* sp. HA, and *Methanosphaera* sp. ISO3-F5. *Methanosarcina* sp. 795 primarily co-occurred with *Methanobrevibacter* sp. AbM23, *Methanoculleus* sp. 25XMc2, *Methanosaeta* sp. HA, *Methanosarcina* sp. 1H1 and *Methanobacterium* sp. MB1. Additionally, *Methanosarcina* sp. 1H1 also frequently co-occurred with *Methanobrevibacter* sp. AbM23, *Methanoculleus* sp. 25XMc2, and *Methanosaeta* sp. HA. Most environmental factors were positively related to the methanogenic communities, except for *Methanobrevibacter ruminantium*, which was negatively correlated with acetate, δ^13^CO_2_, TC, AK, TN, and soil δ^13^C. Overall, soil C—which was strongly, positively related to *Methanosarcina* sp. 795, *Methanobacterium* sp. MB1, *Methanosarcina* sp. 1H1, *Methanosaeta* sp. HA, uncultured *Methanomethylovorans* sp., uncultured *Methanosarcina* sp. and an uncultured methanogen—was the environmental factor that most strongly influenced paddy methanogenic community structures.

## 3. Discussion

### 3.1. Rice Photosynthesis-Derived and Native Soil-Derived Methane Emissions

Addition of organic manure is commonly implemented in rice agriculture to improve soil fertility and recycle agricultural waste, but inevitably enhances CH_4_ production in flooded paddy soil [40,41]. However, the extent to which organic fertilizer increases CH_4_ emissions remains unknown. This study indicates that approximately 49% of CH_4_ emissions from rice paddy soil are derived from rice photosynthesis, and the application of organic fertilizer increases photosynthesis-derived CH_4_ emissions in rice soil. The remainder of the CH_4_ emissions may originate from the native soil or organic fertilizer, and this needs further calculation. In this study, the applied dose of organic fertilizer was same to the local N fertilization of 300 kg·N·ha^−1^. By calculation, it was 10.24 g organic fertilizer, which included 2.36 g C. While, it was 359.67 g C in the topsoil (20 cm). Therefore, the fertilizer C:soil C ratio was 0.66%. If mineralization of the organic fertilizer transformed all of the fertilizer-derived C to CH_4_ (which is unlikely) [38], CH_4_ emissions derived from the organic fertilizer would account for only 0.34% of total CH_4_ emissions at most. Consequently, at least 50.60% of the C sources for CH_4_ emissions must be derived from the native soil. Therefore, the increased CH_4_ emissions observed in organically-fertilized paddy soil can ascribed to enhanced rice photosynthesis and improved soil fertility. 

Previous studies showed that up to 60% of CH_4_ originated from rice photosynthesis [34]. Atmospheric CO_2_ is brought into the rice during photosynthesis and partially utilized by methanogens, especially *Methanocellales* (Rice Cluster I) and *Methanosaetaceae* [35,42]. However, not all soil methanogens can survive in rice roots and, thus, use organic C substrates to produce CH_4_ [34]. The contribution of rice photosynthesis to CH_4_ emissions in paddy fields was found to be influenced by the community abundances and activities of methanogens in the rice roots [14]. Previous ^13^CO_2_ labelling studies showed more than 85% of the ^13^C accumulated in the aerial parts of the plant, approximately 10% was transferred to the roots, less than 2% was transferred to the soil, and only 0.3% was transferred to CH_4_ [43]. At the end of cultivation, we found 86.49–91.38% of the ^13^C was present in the plants, 0.62–1.08% in the soil, and 2.19–2.98% in CH_4_. Thus, rice photosynthesis-derived C is an important source of metabolic substrates for methanogens. In addition, δ^13^CH_4_ and rice biomass were higher in the organic fertilizer treatment than no fertilizer treatment, especially at the tillering stage (Appendix A). This indicates CH_4_ emissions increased with the rice photosynthetic activity [44]. However, C originating from rice photosynthates is just one component of CH_4_ emissions. The C substrate utilization efficiency of methanogens may be a more important factor that determines the increase in CH_4_ emissions from organically fertilized paddy soil.

Apart from the organic fertilizer- and rice photosynthesis-derived CH_4_ emissions, 50.60% of C sources for total CH_4_ emissions were derived from the native soil. This suggests that native or pre-existing soil C became more labile to microbial metabolism after organic fertilizer was applied [38]. The utilization efficiency of soil C substrates or native C sources was previously found to be of pivotal importance to CH_4_ emissions from rice soil [34]. However, the mechanisms by which native C sources transform to active C and finally convert into CH_4_ are unknown and merit further research. Essentially, CH_4_ emissions from paddy fields are the result of a variety of complex soil microbial processes, including CH_4_ production in flooded anaerobic microsites and consumption (oxidation) in aerobic microsites [22,23]. In our previous research, we found soil C, N, potassium, phosphate, and acetate were strongly related to CH_4_ emissions during all stages of rice growth, especially after organic fertilizer was applied [31]. This suggests that the ability of organic fertilizer to increase CH_4_ emissions is not due to the introduction of extra C sources, but instead was due to the alterations to soil properties. Therefore, the increase in CH_4_ emissions induced by the addition of organic fertilizer are primarily due to changes in labile soil characteristics (N, P, K, and acetate) that affect the availability of soil C substrates to methanogens and other microbial communities in paddy soil.

### 3.2. Acetoclastic Methanogens Dominate Rice Soil after Application of Organic Fertilizer

Methanogens occur widely in anaerobic environments because of their unique ability to harvest energy using CO_2_ (electron acceptor) and/or methylated compounds as an energy source [45,46]. Despite the widespread presence of methanogens, global CH_4_ production primarily stems from subsurface sediments that contain a small proportion of the methanogenic population, corresponding to <1% of the total microbiome and a small fraction of archaea [47]. The dynamics of methanogenesis and methanogen community structure in flooded rice fields depend on various parameters, such as soil properties, rice cultivar, fertilization, seasonal stage, crop rotation, and climate change [33]. Analysis of methanogenic community structure and function in paddy fields will provide a better understanding of the ecology of methanogens and their CH_4_ production potential [3,6,23]. This is essential to formulate a strategy to mitigate CH_4_ emissions from rice agriculture. Methanogens are classified into three categories: hydrogenotrophic, acetoclastic, and methylotrophic [11,16]. Acetoclastic and hydrogenotrophic methanogens were the primary physiological groups found in rice fields [48,49]. Acetoclastic methanogens convert acetic acid to CH_4_ and CO_2_ and contain two families, *Methanosarcinaceae* and *Methanosaetaceae* [48]. However, *Methanosarcinaceae* exhibit hydrogenotrophic and acetotrophic nutritional modes [50]. We found *Methanosarcina* dominated the organically fertilized rice soil, with a higher relative abundance in rice root soil than non-root soil. In rice soil, CH_4_ is produced by methanogens inhabiting the rhizosphere and non-rhizosphere bulk soil, as well as the surfaces of the roots [6,34,51]. In contrast to the non-rhizospheric or bulk region, transport of atmospheric O_2_ via the rice aerenchyma to the roots and the release of dissolved organic compounds in the form of root exudates occur in the rhizosphere region [34]. Thus, the microorganisms responsible for the degradation and utilization of substrates in these regions show a heterogenous community composition. Another acetoclastic methanogen, *Methanosaeta*, was the predominant methanogenic community identified across all rice soil treatments in our former field research [31]. The environmental factors affecting methanogenic community dynamics—such as rice plant biomass, soil available nutrients, and CO_2_ and CH_4_ emissions in rice fields—have been previously discussed. Several researchers have proven that soil characteristics control methanogenic community structures [9,10,52]. We found organic fertilization led to expansion of dominant acetoclastic methanogens due to the increased availability of soil nutrients, including soil-available N, P, and K. We assume that production of CH_4_ from decomposition of acetate could be dramatically increased by organic fertilization, and this hypothesis merits further research.

Under the current taxonomic system in the Silva or NCBI databases, *Methanosarcina* sp. 795 and *Methanosarcina* sp. 1H1 had a symbiotic relationship with *Methanobrevibacter* sp. AbM23, *Methanoculleus* sp. 25XMc2, *Methanosaeta* sp. HA, and *Methanobacterium* sp. MB1, as well as some uncultured methanogenic species. Kouzuma et al. [53] found that the mcrA phylogenetic tree constructed for *Methanosarcina* showed *Methanosarcina* sp. 795 was affiliated with *Methanosarcina thermophila* in an enrichment culture from an acetate-fed thermophilic digester in Canada. *Methanosarcina* sp. 795 was also very closely related to *Methanosarcina thermophila* CHTI-55 isolated in France [54]. Close relatives of *Methanosarcina thermophila* have been widely detected in thermophilic anaerobic conditions worldwide. These observations suggest this taxon plays an important ubiquitous role in anaerobic digesters, regardless of geographical location. In this study, *Methanosarcina* sp. 795, as well as *Methanosarcina* sp. 1H1 were found in rice soil. Ijiri et al. [55] cultivated the methanogen 1H1 strain for approximately nine months to study a pure isolation of spherical cells. Phylogenetic analysis of the 16S rRNA gene showed strain 1H1 was closely related to *Methanosarcina mazer*. The *Methanosarcina mazer* isolate could grow on H_2_/CO_2_, acetate, methanol, dimethylamine, and trimethylamine, but not on formate, dimethylsulfide, ethanol, 1-propanol, 2-propanol, cyclopentanol, 1-butanol, or 2-butanol. As shown in Figure 5, *Methanosarcina* sp. 795 and *Methanosarcina* sp. 1H1 had a strong symbiotic relationship. In their role as hydrogenotrophic methanogens, CO_2_ is first reduced and activated to formyl-methanofuran with reduced ferredoxin (Fd_red_) as the electron donor [56]. However, during acetoclastic methanogenesis, acetate is first activated [42], then converted with ATP and coenzyme A (CoA) to acetyl-CoA, which is then split by the CODH/acetyl-CoA synthase complex. The methyl group is transferred to H_4_MTP, which is tetrahydrosarcinapterin (H_4_SPT) in *Methanosarcina*. H_4_SPT is then converted to CH_4_ via a process similar to the CO_2_ reduction pathway (Figure 1). The carbonyl group is oxidized to CO_2_, thus providing the electrons for methyl group reduction. However, cytochrome-containing *Methanosarcina* (including *Methanosarcina acetivorans*, *Methanolobus tindarius*, and *Methanothrix soehngenii*) produce less CH_4_ than hydrogenotrophic methanogens under many environmental conditions [46]. This indicates many *Methanosarcina* would switch to the acetoclastic pathway in the presence of sufficient acetate to produce CH_4_ [48]. However, the functional role of *Methanosarcina* and main active species involved in CH_4_ production are still unknown. Other minor groups could be more active. In the future, the contributions of different methanogenic groups need to be verified.

## 4. Materials and Methods

### 4.1. Experimental Site

The experimental soil was from a typical paddy field located at the Irrigation Technology Research Station (121.12° E, 31.15° N, altitude 5 m) in Qingpu, Shanghai, China. The site experiences a subtropical humid monsoon climate with an annual average daily temperature of 15.8 °C, annual average rainfall of 1300 mm, and 1400 h of sunshine per year. Rotation between winter wheat *cv. Yangmai-5* and summer rice *cv. Huayou-14* was the major cropping system since 2009 [30,32]. The local nitrogen fertilization is 300 kg·N·ha^−1^ for the total paddy season. After rice harvesting in October 2016, we dug about 300 kg soil (0–20 cm) with hoes and shovels from the field. Before pot cultivation, we took 500 g soil for air-drying and passed through a 2 mm sieve to measure the physicochemical characteristics [30]. The soil had a pH of 7.71, EC of 0.11 mS·cm^−1^, bulk density of 1.43 g·cm^−3^, total nitrogen (TN) content of 0.43%, total carbon content (TC) of 1.78%, available phosphorus content of 19.53 mg·kg^−1^, and available potassium content of 84.34 mg·kg^−1^. The fresh soil was directly taken to the greenhouse. Every 20 kg soil was packed into a column-shaped bucket.

### 4.2. Experimental Design

The experiment was conducted with four treatments and three replicates: 1) no fertilizer as a control (CN), 2) no fertilizer + ^13^CO_2_ (CY), 3) organic fertilizer (ON), and 4) organic fertilizer + ^13^CO_2_ (OY). The organic fertilizer was chicken manure with moisture of 23%, TN content of 19 g·kg^−1^, TC content of 202 g·kg^−1^, total phosphorus content of 13 g·kg^−1^, and total potassium content of 29 g·kg^−1^. It was supplied by Sennong Environmental Protection Technology Co. Ltd. (Shanghai, China). Two days before rice transplantation, 10.24 g organic fertilizer were applied to each pot (calculated according to the same dose as the local N fertilization of 300 kg·N·ha^−1^). Afterwards, thirty-day-old rice seedlings (*cv. Huayou-14*) with uniform growth tendency (approximately 15 cm tall) were transplanted to the fresh soil. Every pot had three clusters of rice seedlings (a cluster = 3 rice seedlings). The paddies were flooded with 5 cm water layer during the whole pot experiment. The greenhouse was maintained at 20–35 °C and a daytime light intensity of 35 kLux. After rice transplanting, the pots were placed into the 490 L incubation chamber (Appendix A). Generally, the rice growth is classified into seven stages: regreening, tillering, jointing, booting, heading, filling, and maturing. Paddy CH_4_ fluxes have been shown to be significantly higher from the regreening to booting stages than others [30,31,57]. In this experiment, rice was cultivated from the regreening to booting stage, lasting for 70 days. 

Isotopic labelling was generated inside the chamber via the reaction between the hydrochloric acid (HCl) and Na_2_^13^CO_3_ (99.80% purity; Cambridge Isotope Laboratories Inc., Tewksbury, MA, USA). The Na_2_^12^CO_3_ was applied as the control. Beforehand, 10 g of Na_2_^13^CO_3_ or Na_2_^12^CO_3_ were put inside the chamber. The 200 mL 1 mM HCl solution was then gently added to the chamber through a tube (Appendix A). The chamber was kept sealed during the labelling experiment. The specific experimental stages were listed below:

(1) Rice regreening stage (first stage, 9 d). Twelve pots containing rice plants with homogeneous growth were selected for the first stage of ^13^C labelling. The first day of labelling was set as day 1 and labelling ended on day 10.

(2) Rice tillering stage (second stage, 27 d). On day 11, new 12 pots containing rice plants with homogeneous growth congruent with the growth trends of the previous stage were selected for the second stage of labelling. Labelling lasted from day 11 until day 37.

(3) Rice booting stage (third stage, 33 d). On day 33, 12 pots containing rice plants with homogeneous growth congruent with the growth trends of the previous stage were selected for the third stage of labelling, which lasted from day 38 to day 70.

### 4.3. CH_4_ and CO_2_ Sampling

Gas samples were collected from the incubation chamber at 0, 6, 12, 24, 36, 48, and 72 h after the initiation of labelling, then once every five days (at 9:00–9:30) until the end of labelling. Measurements included δ^13^CH_4_, δ^13^CO_2_, total CH_4_, and total CO_2_. The concentrations of CH_4_ and CO_2_ were analyzed at the Analytical Platform of Shanghai Jiao Tong University using an Agilent 7820A gas chromatograph (Agilent, Santa Clare, CA, USA). Gas fluxes and emissions were calculated as described by Yuan et al. [30]. Additionally, δ^13^CH_4_ (accuracy δ^13^C ± < 0.5‰) and δ^13^CO_2_ (accuracy *δ^13^C* ± < 0.3‰) were analyzed by Huakejingxin Technology Co., Ltd. (Shenzhou, China) using a Delta V^TM^ Isotope Ratio Mass Spectrometer (Thermo Fisher Scientific, Grand Island, NY, USA).

(1) δ^13^C was calculated using Equation
:δ^13^C (‰) = 1000 × (*R_sample_* − *R_standard_*)/*R_standard_*(3)
where δ^13^C (‰) is the abundance of the isotope composition, *R_sample_* is the *^13^C_sample_*/*^12^C_sample_* for each sample, and *R_standard_* is the *^13^C_standard_*/*^12^C_standard_* of the international common C isotope analysis standard (usually the ^13^C/^12^C ratio of Pee Dee Belemnite [PDB]).

(2) Daily accumulation of δ^13^C was calculated using Equation (4):*V* (‰·d^−1^) = (δ^13^C_2_ − δ^13^C_1_)/*t*(4)
where δ^13^C_2_ and δ^13^C_1_ are the δ^13^C values of the first and last day of each stage, respectively, and *t* is number of experimental days for each stage. 

(3) δ^13^CH_4_ per rice biomass was calculated using Equation (5):Rice δ^13^CH_4_ (‰·g^−1^) = Final δ^13^C/*S_chamber_*/*W_rice_*(5)
where rice δ^13^CH_4_ (‰·g^−1^) is δ^13^CH_4_ per unit rice biomass; final δ^13^C is the ultimate δ^13^C for the organic fertilizer and no fertilizer treatments in the rice soil; *S_chamber_* is the area of the chamber, (0.49 m^2^); *W_rice_* (g·m^−2^) is the weight of the rice biomass.

(4) Contribution coefficients for δ^13^CH_4_ responding to rice plant or soil were calculated using Equations (6)–(9):*y_r_* = *a_r_x* + *b_r_*(6)
*y_s_* = *a_s_x* + *b_s_*(7)
*K_r_* = 1/*a_r_*(8)
*K_s_* = 1/*a_s_*(9)
where *y_r_* is the daily rice plant δ^13^C (‰·d^−1^); *y_s_* is daily rice soil δ^13^C (‰·d^−1^); *x* is daily δ^13^CH_4_ (‰·d^−1^); *K_r_* is taken as the contribution coefficient of rice C accumulation to CH_4_ emission; *K_s_* is taken as the contribution coefficient of rice soil C fixation to CH_4_ emission.

### 4.4. Plant and Soil Collection

On the last day of each experimental labelling stage, the rice plants and soil were collected. Plant samples (grains, leaves, shoots, and roots) were oven-dried at 60 °C. Soil samples were homogenized, air-dried, crushed and passed through a sieve (<2 mm). Soil texture was determined using the Bouyoucos hydrometer method. To assess pH and electrical conductivity (EC), soil samples were vigorously mixed with deionized water at a 1:2.5 solid:water ratio for 20 min and assessed using digital meters (Elico Model LI-120, Elico Pvt. Ltd., Hyderabad, India). Organic C was determined using the Walkley–Black method. The δ^13^C and total C content of the plant and soil samples were determined at the Instrumental Analysis Center of Shanghai Jiao Tong University using an elemental isotope mass spectrometry analyzer (Vario EL III/Isoprime, Agilent, Santa Clara, CA, USA). 

### 4.5. DNA Extraction and MiSeq Sequencing of 16S rRNA

At the tillering stage, the rice rhizosphere (root) and non-rhizosphere (non-root) soils were collected separately and labelled as C_non-root, C_root, O_non-root, and O_root. All soil samples were placed into sterile sealable plastic bags and stored at −70 °C. Total soil DNA was extracted from 0.5 g rhizosphere soil using BioFast Soil Genomic DNA Extraction kits (Tiangen Co., Ltd., Beijing, China). The extracted DNA was eluted in 100 μL Elution Buffer and stored at −20 °C until the MiSeq Illumina analyses. Amplicon library preparation and Illumina MiSeq sequencing were performed by Majorbio Bio-Pharm Technology Co., Ltd. (Shanghai, China).

Amplification of archaeal 16S rRNA gene fragments was conducted using the primers 1106F/1378R (F: TTWAGTCAGGCAACGAGC and R: TGTGCAAGGAGCAGGGAC) [58] targeting the V4 region (280 bp product). In addition to the 16S target-specific sequence, these primers also contain adaptor sequences to allow uniform amplification of a library with high complexity to enable downstream NGS sequencing on an Illumina MiSEquation (PCR was conducted using a TransGen AP221-02 reaction kit (TransStart Fastpfu DNA Polymerase, Majorbio Inc., Shanghai, China) on an ABI GeneAMP^®^ 9700 (Thermo Fisher Scientific, Grand Island, NY, USA) with initial denaturation at 95 °C for 90 s, followed by 35 cycles of 95 °C for 30 s, 55 °C for 30 s and 72 °C for 90 s, and a final extension at 72 °C for 6 min. Three replicates were assessed for each sample and the triplicate PCR products for each sample were mixed and separated on 2% agarose gels to recover the PCR products. The amplicons were purified using an AxyPrepDNA gel recovery kit (AXYGEN Co., Ltd., Tewksbury MA, USA) and quantified using the QuantiFluor™-ST Blue system (Promega Co., Ltd. Madison, WI, USA). Finally, the samples were pooled at equimolar concentrations (>2 nM) and sequenced on separate MiSeq runs using a 2 × 300 bp paired end protocol. Finally, the TruSeq^TM^ DNA Sample Prep Kit (Majorbio Inc., Shanghai, China) was used for MiSeq library construction.

The paired end reads obtained from the MiSeq sequencing were first spliced according to the overlap relationship, and quality control and filtration of sequence quality were conducted to distinguish the sample reads, followed by cluster and taxonomy analyses. The double-ended FASTQ sequences were filtered using the sliding window method, and Trimmomatic and FLASH software programs were used to connect the sequences that passed the quality filter. The FASTA and qual files from Mothur were converted to FASTQ format using USEARCH (Version 7.1) [59]. Downstream processing and operational taxonomy unit (OTU) picking were performed using UPARSE [60]. Barcodes and primers from the merged sequences were removed. Sequence reads < 200 bp were discarded. After dereplication and abundance sorting keeping singletons, sequences with a minimum of 97% similarity were clustered into OTUs using the average neighbor algorithm. Taxonomy was assigned using the Silva database with PyNAST at an 80% confidence threshold [52,61]. The sequence data are available from the NCBI Nucleotide Archive database under project no. PRJNA450890 and accession number SRP142386.

### 4.6. Statistical Analysis

The experiment was conducted with three replicates for each treatment. SPSS 22.0 software (IBM, NY, USA) was used to analyze the differences between treatments using one-way analysis of variance (ANOVA) and the Pearson correlation test. Figures were generated using Origin 8.0 (OriginLab Co., Northampton, MA, USA). The Venn diagram was generated using the ‘Vennerable’ package in R (MathSoft Co., Needham, MA, USA). QIIME was used to calculate the weighted pairwise Uni-Frac distances for community comparisons, which were visualized using non-metric multidimensional scaling (NMDS) plots in PRIMER v6 [61]. Clustering was performed using the ‘complete linkage’ method, and the heatmap based on clustering was constructed with the package hclust in R [62]. Redundancy analysis (RDA) was performed to summarize the variations in methanogenic communities that could potentially be explained by the variables (treatments, CH_4_ efflux and all physicochemical parameters) using CANOCO 4.5 software (Microcomputer Power, Ithaca, NY, USA) [31]. Network analysis was conducted on the methanogenic OTUs and soil properties using the maximal information coefficient (MIC) implanted in python. The MIC is a highly informative score that reveals the strength of linear and non-linear associations among variables. Only 24 OTUs belonging to the methanogenic community were identified at the species level. Therefore, all the relationships were used for network construction in Cytoscape v.3.2.1 [63]. Network topological characteristics were calculated using the NetworkAnalyzer tool in Cytoscape.

## 5. Conclusions

C sources derived from rice photosynthesis and native soil are the two-primary pathways for CH_4_ emissions from paddy soil. Daily δ^13^CH_4_ accumulation was positively dependent on daily rice plant δ^13^C and soil δ^13^C. Moreover, the increase in CH_4_ emissions observed in organically fertilized soil was sensitive to soil C fixation. After 16S rRNA sequencing, we found organic fertilization led to significantly different microbial community structures in both the root and non-root areas of rice soil. Acetoclastic *Methanosarcina* was the dominant methanogenic community in rice soil overall. *Methanosarcina* sp. 795 and *Methanosarcina* sp. 1H1 co-occurred with *Methanobrevibacter* sp. AbM23, *Methanoculleus* sp. 25XMc2, *Methanosaeta* sp. HA, and *Methanobacterium* sp. MB1, as well as other unknown and uncultured methanogens. The increase of CH_4_ emissions and variance of methanogenic community structure observed after addition of organic fertilizer to rice soil were primarily due to the increases in labile soil C, N, P, K, and acetate.

## Figures and Tables

**Figure 1 ijms-20-01586-f001:**
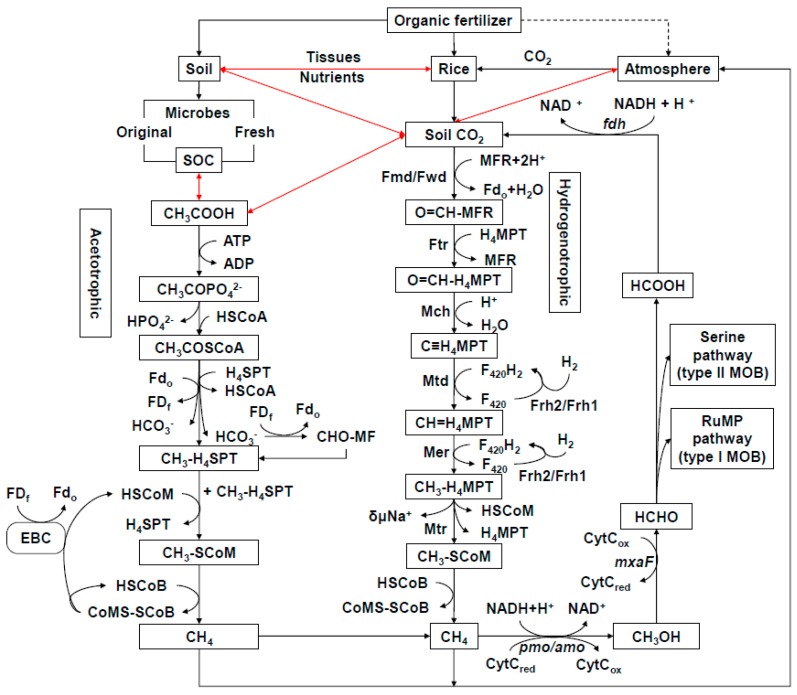
Effects of organic fertilizer on the pathway of CH_4_ emission from rice soil. The black lines indicate a transformed direction. The red lines represent bidirections between two matters. The acetoclastic methanogenic pathway was referenced from Soo et al. [17]. SOC, soil organic carbon. CH_3_COOH, acetate. ATP, adenosine triphosphate. ADP, adenosine diphosphate. CH_3_COPO_4_^2-^, acetyl phosphate. HPO_4_^2-^, hydrogen phosphate ion. HSCoA, coenzyme A. CH_3_COSCoA, acetyl-CoA. Fd_o_, oxidized ferredoxin. Fd_f_, reduced ferredoxin. H_4_SPT: 5,6,7,8-tetrahydrosarcinapterin. HCO_3_^-^, bicarbonate ion. CHO-MF, formyl methanofuran. CH_3_-H_4_SPT, methyl-H_4_SPT. HSCoM, coenzyme M. CH_3_-SCoM, methyl-coenzyme M. HSCoB, coenzyme B. CoMS-SCoB, heterodisulfide of coenzyme M and coenzyme B. EBC, electron bifurcating complex. The hydrogenotrophic methanogenic pathway was referenced from Liu et al. [20]. Fmd/Fwd, CHO-MFR dehydrogenase. MFR, methanofuran. Ftr, formyltransferase. H_4_MPT, tetrahydromethanopterin. Mch, methenyl-H_4_MPT cyclohydrolase. Mtd, methylene-H_4_MPT dehydrogenase. F420, coenzyme F420. Frh, reducing hydrogenase. Mer, methylene-H_4_MPT reductase. Na, electrochemical sodium ion potential. Mtr, methyl-H_4_MPT methyltransferase. The methanotrophic pathway was referenced from Cai et al. [22]. NADH/NAD^+^: Nicotinamide adenine dinucleotide. CH_3_OH: Methanol. HCHO: methanal. HCOOH: methanoic acid. CytC_red_: Cytochrome C reduction. CytC_ox_: Cytochrome C oxidation. pmo: methane monooxygenase (particulate). amo: ammonia oxidase. mxaF: methanol dehydrogenase. fdh: formate dehydrogenase. RuMP pathway (type I MOB): phosphoribulose pathway (methanotroph type I). Serine pathway (type II MOB): Serine pathway (methanotroph type II).

**Figure 2 ijms-20-01586-f002:**
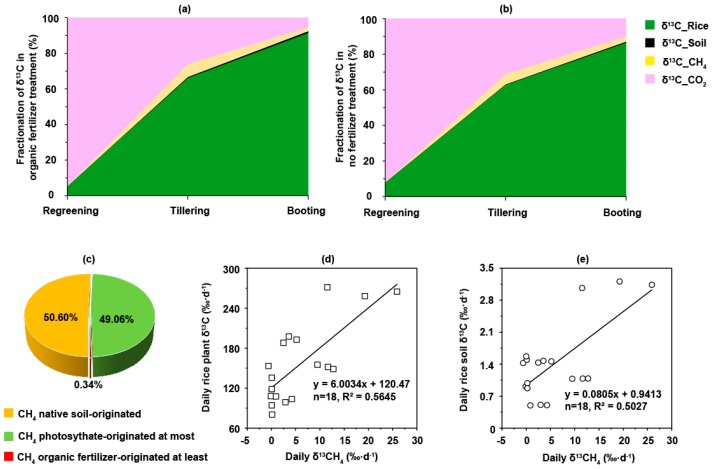
Relative abundance of δ^13^C from cultured rice soil. (**a**,**b**) are the distributions of δ^13^C in the CO_2_, CH_4_, rice plant and soil for the average values of organic fertilizer and no fertilizer treatment. (**c**) shows the total carbon sources for the CH_4_ emissions in the cultivated rice soil. (**d**,**e**) are the linear regressions for the daily δ^13^CH_4_ to the daily rice plant δ^13^C and soil δ^13^C. Meanwhile, CH_4_ fluxes, δ^13^CH_4_ and rice biomass are shown in Appendix A, CO_2_, CH_4_ emissions and ^13^C atom percentage (%) in Appendix A, and soil and rice plant δ^13^C in Appendix A. Results indicated that approximately 49.06% of the carbon in the CH_4_ was derived from rice photosynthates. Therefore, the rest of the carbon sources would be approximately 50.95%, which should come from organic fertilizer or native soil. The average carbon and nitrogen of the organic manure was 23.05% and 2.07%. We knew that the pure nitrogen was applied at 300 kg·N·ha^−1^. By calculation, the carbon from the organic fertilizer was applied at 2.36 g in the soil. Alternatively, the soil total carbon and bulk density were 1.78% and 1.43 g·cm^−3^, respectively. The total amount of carbon in the topsoil (20 cm) was approximately 359.67 g. Therefore, the carbon ratio of the organic fertilizer and original soil would be 0.66%. Consequently, the carbon source for the CH_4_ emissions derived from the organic fertilizers would be 0.34% at most, and 50.60% were from the soil at least. This was how we calculated the carbon sources of CH_4_ in the rice soil after the organic fertilizer was applied.

**Figure 3 ijms-20-01586-f003:**
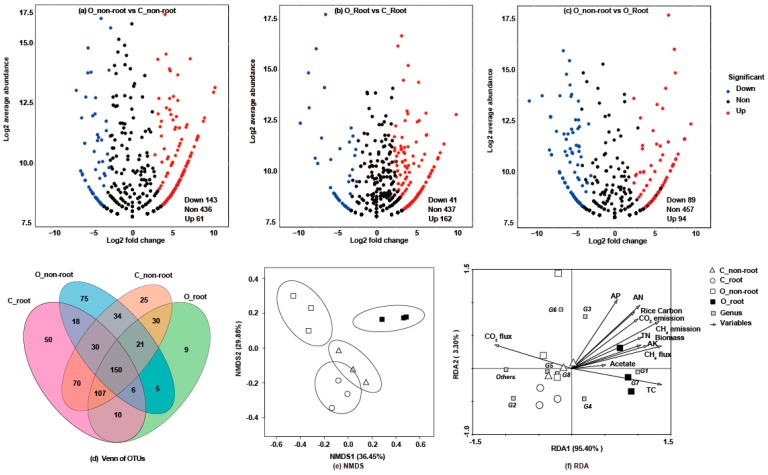
Profiles of soil samples from 16S rRNA sequencing. O_non-root, OTUs from the rice non-root soil of organic fertilizer treatment. C_non-root, OTUs from the rice non-root soil of no fertilizer treatment. O_root, OTUs from the rice root soil of organic fertilizer treatment. C_root, OTUs from the rice root soil of no fertilizer treatment. After sequencing, we obtained 645 series of OTUs. The volcano plots (**a**–**c**) illustrated that the OTUs were significantly enriched (red) and depleted (blue) for each sample. Each point represented an individual OTU, and the *X*-axis indicated the abundance fold change between two soil samples. Using the OTU abundance from the C_non-root soil or the C_root soil as the control and an adjusted *p*-value cutoff of 0.1, “down OTUs (Down)”, “up OTUs (up)” and “none down or up OTUs (None)” specifically represent OTUs that increase and decrease significantly in relative abundance from different soil samples, respectively. The Venn plot (**d**) involved four sets of ellipses: O_non-root, C_non-root, O_root, and C_root. Numbers within the diagrams indicated whether the OTUs were shared among treatments or unique to one given treatment. In addition, the Nonmetric Multidimensional Scaling (NMDS **e**) analysis was performed to reveal the distances within the samples. The NMDS analysis is not affected by numerical sample distance and only considers the size of the relationship between each other. The RDA plot (**f**) was to analyze the correlations between the methanogenic community structure and variables. CO_2_ flux, the CO_2_ flux of the chamber in the whole system. AP, soil available phosphate. AN, soil alkali-hydrolyzable nitrogen. Rice carbon, the total carbon for rice plant. CO_2_ emission, the CO_2_ emission of the chamber in the whole system, calculated by the sum of CO_2_ flux. TN, soil total nitrogen. Biomass, the dry weight of rice plant. AK, soil available potassium. CH_4_ flux, the CH_4_ flux of the chamber in the whole system. Acetate, soil acetate. TC, soil total carbon. δ^13^CH_4_, the δ^13^CH_4_ abundance in the whole system. Soil δ^13^C, soil δ^13^C abundance. G1, *Methanosarcina*; G2, *Methanolobus*; G3, *Methanomethylovorans*; G4, *Methanoculleus*; G5, *Methanobacterium*; G6, *Methanobrevibacter*; G7, *Methanosphaera*; G8, *Methanosaeta*.

**Figure 4 ijms-20-01586-f004:**
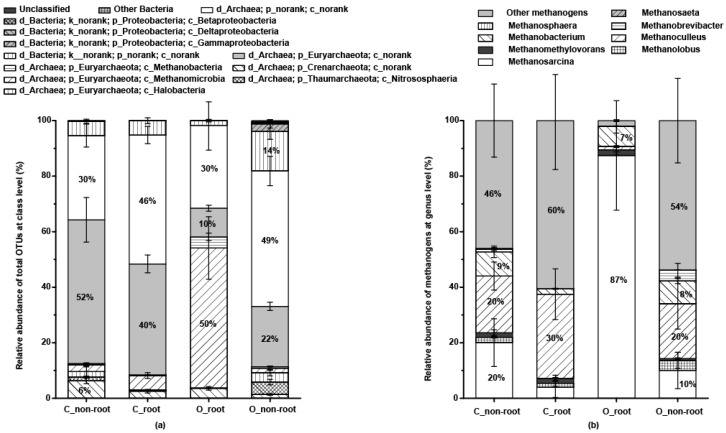
Relative abundance of the whole OTUs and methanogenic communities. (**a**) is the methanogenic communities classified at the taxonomic class level; (**b**) is the methanogenic communities classified at the taxonomic genus level. O_non-root, OTUs from the rice non-root soil of organic fertilizer treatment. C_non-root, OTUs from the rice non-root soil of no fertilizer treatment. O_ root, OTUs from the rice root soil of organic fertilizer treatment. C_ root, OTUs from the rice root soil of no fertilizer treatment. In the taxonomy database, due to the large data of microbial communities, there will be many taxonomic lineages in the intermediate level without a scientific name, labelled “no rank”. In addition, the taxonomic comparison is based on the selection of the confidence threshold. After BLAST, some hierarchical classification would receive relatively low scores, marked as “unclassified”.

**Figure 5 ijms-20-01586-f005:**
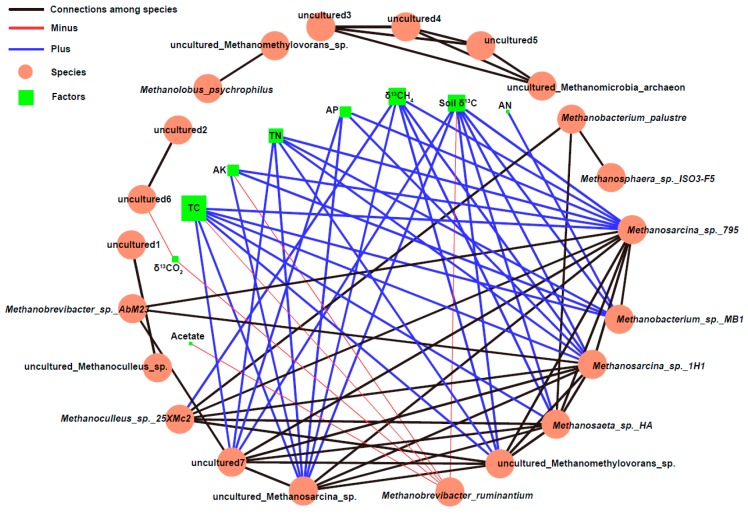
Co-occurrence networks. Acetate, soil acetate. δ^13^CO_2_, the δ^13^CO_2_ abundance in the whole system. TC, soil total carbon. AK, soil available potassium. TN, soil total nitrogen. AP, soil available phosphate. δ^13^CH_4_, the δ^13^CH_4_ abundance in the whole system. Soil δ^13^C, soil δ^13^C abundance. AN, soil alkali-hydrolyzable nitrogen. The red lines, negative correlations. The blue lines, positive correlations. The black lines, correlations among methanogens. The thickness of lines, the value of correlations. The red circles, methanogenic species. The green square, factors. *Methanobacterium palustre*, Taxonomy ID 2171 in NCBI. *Methanosphaera* sp. ISO3-F5, taxonomy ID 1,452,353 in NCBI. *Methanosarcina* sp. 795, taxonomy ID 1,653,396 in NCBI. *Methanobacterium* sp. MB1, taxonomy ID 1,379,702 in NCBI. *Methanosarcina* sp. 1H1, taxonomy ID 1,882,224 in NCBI. *Methanosaeta* sp. HA, taxonomy ID 1,562,961 in NCBI. *Methanobrevibacter ruminantium*, no taxonomy ID in NCBI. *Methanoculleus* sp. 25XMc2, taxonomy ID 1,898,331 in NCBI. *Methanobrevibacter* sp. AbM23, taxonomy ID 1,452,345 in NCBI. *Methanolobus psychrophilus*, taxonomy ID 420,950 in NCBI.

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
