# Peer review of "Three-Source Partitioning of Methane Emissions from Paddy Soil: Linkage to Methanogenic Community Structure"

_ijms, 2019, doi:10.3390/ijms20071586_

Round 1

Reviewer 1 Report

Interesting work to know more on the different methanogens  and what about CH4 in the atmosphere

Some corrections to do:

lines338-368: the names of methanogens must be in italics

line 371: precise how was done the collect, anerobiosis? and how were conserve the samples?

Be more precise for the culture conditions.

Author Response

Dear Reviewer:

Many thanks for reviewing our manuscript and giving our positive comments.

The specific modifications are listed below:

C1: lines338-368: the names of methanogens must be in italics

Answer: Thank you for your careful observation. The names of methanogens were all rewritten in italics.

C2: line 371: precise how was done the collect, anerobiosis? and how were conserve the samples? Be more precise for the culture conditions.

Answer: Thank you for your suggestion. The 4.1 and 4.2 parts has been rewritten to make the experiment more precise. The first paragraph was about the soil collection site and its physiochemical characteristics. The second paragraph described the rice cultivation, conditions, facilities and time, etc.

Reviewer 2 Report

The current study examined the source partitioning of methane emission from paddy soil. In general, the manuscript is well-written and addresses a critical component of C-cycling in paddy soils. My major concern is the low number of replicates used in this experiment. Add more information on the variability of the soils used in this experiment and indicate how the soil variability was addressed in the statistical analyses. The novelty of the study needs to be highlighted more. For example, the microbial community shift under irrigated paddy soil has been studied in several other previous studies. More background information is needed on what methanogens were expected to increase under organic fertilizer. Use of different statistical methods (e.g. NMDS vs RDA and Co-occurrence network) need to be justified as well. A few minor points below:

Abstract

L24: change ‘root soil’ to ‘rhizosphere’

Introduction:

L70-71: Explain more here; what do you mean by reserved in the soil? Not uptaken by the plants?

L72: complex metabolic pathways such as?

L112: Are you indicating to microcosm in Pump and Conrad (2014)?

Discussion

L263: what was the application rate by area basis? 

Author Response

Dear Reviewer:

Many thanks for reviewing our manuscript and giving our valuable suggestions.

The specific comments (C) and answers (A) are listed below:

C1: My major concern is the low number of replicates used in this experiment.

A1: Thank you for your kind concern. We designed three replicates for each treatment, and every replicate was sampled for thrice each (including air, soil, and microbial DNA).

C2: Add more information on the variability of the soils used in this experiment and indicate how the soil variability was addressed in the statistical analyses.

A2: Thank you for this suggestion. This experiment was designed based on our former study (Yuan, J., et al., 2018. Effects of different fertilizers on methane emissions and methanogenic community structures in paddy rhizosphere soil. Science of The Total Environment. 627, 770-781.). Four treatments, chemical fertilizer (CT), organic fertilizer (OT), mixed with chemical and organic fertilizer (MT), and no fertilizer (ctrl), were involved in the paddy field. We found CH4 emissions were completely higher in OT (145.31 kg ha−1) than that in MT (84.62 kg ha−1), CT (77.88 kg ha−1) or ctrl (32.19 kg ha−1). Moreover, we found that soil nitrogen, carbon, phosphate, potassium and acetate were the main key environmental factors associated with CH4 emissions and dominant methanogenic communities. To make further investigation, we designed this pot experiment and choosing those representative soil variables for the next statistical analysis. The specific contents of soil variables were provided in the supplementary. The objective of this study was going to find the main carbon sources for CH4 emissions and the fundamental environmental factor with the application of organic fertilizer in paddy soil.

C3: The novelty of the study needs to be highlighted more. For example, the microbial community shift under irrigated paddy soil has been studied in several other previous studies.

A3: Thank you for this valuable advice. We made some amends in the abstract (line 12 – line 17) and introduction (line 107 to line 126, and line 143 to line 146).

C4: More background information is needed on what methanogens were expected to increase under organic fertilizer.

A4: Thank you for your useful advice. We add some information in Line 112 to line 120.

C5: Use of different statistical methods (e.g. NMDS vs RDA and Co-occurrence network) need to be justified as well.

A5: Thank you for your question. The non-metric multidimensional scaling (NMDS) was applied to calculate the weighted pairwise Uni-Frac distances for community comparisons. Redundancy analysis (RDA) was performed to summarize the variations in methanogenic communities that could potentially be explained by the variables by CANOCO 4.5 software. The co-occurrence network was conducted on the methanogenic OTUs and soil properties using the maximal information coefficient in Python software. The results were calculated and visualized in Cytoscape v.3.2.1. The specific descriptions for NMDS, RDA and network were shown in 4.6 statistical analysis.

C6: L24: change ‘root soil’ to ‘rhizosphere’

A6: Thank you for your careful observation. The ‘root soil’ has been changed to ‘rhizosphere’.

C7: L70-71: Explain more here; what do you mean by reserved in the soil? Not uptaken by the plants?

A7: Thank you for your question. The description has been corrected.

C8: L72: complex metabolic pathways such as?

A8: Thank you for your question. The description has been corrected.

C9: L112: Are you indicating to microcosm in Pump and Conrad (2014)?

A9: Thank you for your question. This sentences have been removed.

C10: L263: what was the application rate by area basis?

A10: Thank you for your question. We explained more here. The specific description was shown in the method. A total amount of 10.24 g organic fertilizer was applied to each pot. This was calculated according to the same dose as the local N fertilization of 300 kg·N·ha-1. The organic fertilizer was chicken manure with moisture of 23%, TN content of 19 g·kg-1, TC content of 202 g·kg-1, total phosphorus content of 13 g·kg-1, and total potassium content of 29 g·kg-1. The size of the cultivation pot was 30 cm wide (diameter) and 30 cm high.